# Ternary MobileNets via Per-Layer Hybrid Filter Banks

## Abstract

MobileNets family of computer vision neural networks have fueled tremendous progress in the design and organization of resource-efficient architectures in recent years. New applications with stringent real-time requirements in highly constrained devices require further compression of MobileNets-like already compute-efficient networks. Model quantization is a widely used technique to compress and accelerate neural network inference and prior works have quantized MobileNets to $4-6$ bits albeit with a modest to significant drop in accuracy. While quantization to sub-byte values (i.e. precision $\leq 8$ bits) has been valuable, even further quantization of MobileNets to binary or ternary values is necessary to realize significant energy savings and possibly runtime speedups on specialized hardware, such as ASICs and FPGAs. Under the key observation that convolutional filters at each layer of a deep neural network may respond differently to ternary quantization, we propose a novel quantization method that generates per-layer hybrid filter banks consisting of full-precision and ternary weight filters for MobileNets. The layer-wise hybrid filter banks essentially combine the strengths of full-precision and ternary weight filters to derive a compact, energy-efficient architecture for MobileNets. Using this proposed quantization method, we quantized a substantial portion of weight filters of MobileNets to ternary values resulting in $27.98\%$ savings in energy, and a $51.07\%$ reduction in the model size, while achieving comparable accuracy and no degradation in throughput on specialized hardware in comparison to the baseline full-precision MobileNets.

## 1 Introduction

Deeper and wider convolutional neural networks (CNNs) has led to outstanding predictive performance in many machine learning tasks, such as image classification (He et al. (2016); Krizhevsky et al. (2012)), object detection (Redmon et al. (2016); Ren et al. (2015)), and semantic segmentation (Chen et al. (2018); Long et al. (2015)). However, the large model size and corresponding computational inefficiency of these networks often make it infeasible to run many real-time machine learning applications on resource-constrained mobile and embedded hardware, such as smartphones, AR/VR devices etc. To enable this computation and size compression of CNN models, one particularly effective approach has been the use of resource-efficient MobileNets architecture. MobileNets introduces depthwise-separable (DS) convolution as an efficient alternative to the standard 3-D convolution operation.While MobileNets architecture has been transformative, even further compression of MobileNets is valuable in order to make a wider range of applications available on constrained platforms (Gope et al. (2019)).

Model quantization has been a popular technique to facilitate that. Quantizing the weights of MobileNets to binary (-1,1) or ternary (-1,0,1) values in particular has the potential to achieve significant improvement in energy savings and possibly overall throughput especially on custom hardware, such as ASICs and FPGAs while reducing the resultant model size considerably. This is attributed to the replacement of multiplications by additions in binary- and ternary-weight networks. Multipliers occupy considerably more area on chip than adders (Li & Liu (2016)), and consume significantly more energy than addition operations (Horowitz (2014); Andri et al. (2018)). A specialized hardware can therefore trade off multiplications against additions and potentially accommodate considerably more adders than multipliers to achieve a high throughput and significant savings in energy for binary- and ternary-weight networks.

However, prior approaches to binary and ternary quantization (Rastegari et al. (2016); Alemdar et al. (2016); Li & Liu (2016); Tschannen et al. (2018)) incur significant drop in prediction accuracy for MobileNets. Recent work on StrassenNets (Tschannen et al. (2018)) presents a more mathematically profound way to approximate matrix multiplication computation (and, in turn, convolutions) using mostly ternary weights and a few full-precision weights. It essentially exploits Strassen's algorithm to approximate a matrix multiplication of a weight matrix with feature maps, where the elements of the product matrix are generated by different combination of few intermediate terms through additions. Computation of each of the intermediate terms requires a multiplication along with combination of different elements of weights and feature maps through additions. The number of intermediate terms (also called *hidden layer width*) in StrassenNets therefore determines the addition and multiplication budget of a convolutional layer and in turn decides the approximation error of the corresponding convolution operation. While the results in (Tschannen et al. (2018)) using StrassenNets demonstrates no loss in predictive performance when compared to full-precision models for few networks, the effectiveness of StrassenNets is quite variable, however, depending on the neural network architecture. We observe, for example, that while *strassenifying* is effective in reducing the model size of DS convolutional layers, this might come with a prohibitive increase in the number of addition operations, reducing the energy efficiency of neural network inference.

The exorbitant increase in additions primarily stems from the use of wide hidden layers for closely approximating each convolutional filter in a network layer. While this might be required for some of the convolutional filters in a layer, our observations indicate that all filters may not require wide strassenified hidden layers. As different filters in a network layer tend to capture different features, they may respond differently to ternary quantization, and, in turn, to strassenified convolution with a specific hidden layer units. Some filters can be harder to approximate using ternary bits than others, and have larger impact on the model accuracy loss. Furthermore, given a constrained hidden layer budget for StrassenNets, a group of filters extracting fairly similar features at a layer may respond favorably to ternary quantization, while other filters of the layer extracting significantly different features from those may not.

Guided by these insights, we propose a layer-wise hybrid filter banks for the MobileNets architecture capable of giving start-of-the-art accuracy levels, while requiring a fraction of the model size and considerably fewer MAC and multiplication operations per inference. The end-to-end learning of hybrid filter banks makes this possible by keeping precision critical convolutional filters in full-precision values and strassenifying quantization tolerant filters only to ternary values. *The filters that are most sensitive to quantization errors perform traditional convolutions with input feature maps, whereas ternary quantization tolerant filters can perform strassenified convolutions using narrow hidden layers.* We apply this proposed quantization scheme to the state-of-the-art MobileNets-V1 architecture. The hybrid filter banks for MobileNets achieves a $46.4\%$ reduction in multiplications, and a $51.07\%$ reduction in model size while incurring modest increase in additions. This translates into a $27.98\%$ savings in energy required per inference while ensuring no degradation in throughput on a DNN hardware accelerator consisting of both MAC and adders when compared to the execution of baseline MobileNets on a MAC-only hardware accelerator. The hybrid filter banks accomplishes this with a very minimal loss in accuracy of $0.51\%$. *To the best of our knowledge, the hybrid filter banks proposed in this work is a first step towards quantizing the already compute-efficient MobileNets architecture to ternary values with a negligible loss in accuracy on a large-scale dataset, such as ImageNet.*

The remainder of the paper is organized as follows. Section 2 elaborates on the incentives behind the use of per-layer hybrid filter banks for the MobileNets architecture and provides a brief overview of current quantization algorithms along with our observations of applying them to the MobileNets architecture. Failing to find a good balance between accuracy and computation costs shifts our focus towards designing layer-wise hybrid filter banks for MobileNets. Section 3 describes our hybrid filter banks. Section 4 presents results. Section 5 compares hybrid filter banks against prior works and Section 6 concludes the paper.

## 2 MODEL QUANTIZATION LIMITATIONS FOR MOBILENETS

Quantization is an extremely popular approach to make DNNs, in particular convolutional neural networks (CNNs), less resource demanding. This section briefly reviews the important existing

works on ternary quantization, which we focus on in this paper, and illustrates their limitations to motivate the development of per-layer hybrid filter banks for quantizing MobileNets to ternary values.

## 2.1 Ternary Quantization of Weights

In order to observe the impact of ternary quantization (Courbariaux et al. (2015); Rastegari et al. (2016); Lin et al. (2017); Cai et al. (2017); Li & Liu (2016); Zhu et al. (2016); Zhou et al. (2016)), we apply the ternary weight quantization method from (Li & Liu (2016)) over the baseline MobileNets-V1 architecture. It approximates a full-precision weight $W^{fp}$ by a ternary-valued $W^t$ and a scaling factor such that $W^{fp} \approx$ scaling factor $* W^t$. Ternary quantization of the weights of MobileNets achieves substantial reduction in model size but at the cost of significant drop (by $9.66\%$, see Table 1) in predictive performance when compared to the full-precision model. Any increase in the size of the MobileNets architecture to recover the accuracy loss while using ternary quantization will lead to a significant increase in the number of addition operations. Recent work on Strassen-Nets (Tschannen et al. (2018)), which we describe next, has shown the potential to achieve near state-of-the-art accuracy for a number of deep CNNs while maintaining acceptable increase in addition operations.

## 2.2 StrassenNets

Given two $2 \times 2$ matrices, Strassen's matrix multiplication algorithm computes their product using 7 multiplications instead of the 8 required with a naïve implementation of matrix multiplication. It essentially converts the matrix multiplication operation to a 2-layer sum-product network (SPN) computation as shown below:

$$vec(C) = W_c[(W_b vec(B)) \odot (W_a vec(A))] \tag{1}$$

$W_a, W_b \in K^{r \times n^2}$ and $W_c \in K^{n^2 \times r}$ are ternary matrices with $K \in \{-1, 0, 1\}$, $vec(A)$ and $vec(B)$ are the vectorization of the two input square matrices $A$, $B \in R^{n \times n}$; and $vec(C)$ represents the vectorized form of the product $A \times B$. $\odot$ denotes the element-wise product. The $(W_b vec(B))$ and $(W_a vec(A))$ of the SPN compute $r$ intermediate factors each from additions, and/or subtractions of elements of $A$ and $B$ realized by the two associated ternary matrices $W_a$ and $W_b$ respectively. The two generated $r$-length intermediate factors are then element-wise multiplied to produce the $r$-length $(W_b vec(B)) \odot (W_a vec(A))$. The outmost ternary matrix $W_c$ later combines the $r$ elements of the product $(W_b vec(B)) \odot (W_a vec(A))$ in different ways to generate the vectorized form of product matrix $C$. Therefore, the width of the hidden layer of the SPN, $r$, decides the number of multiplications required for the Strassen's matrix multiplication algorithm. For example, given two $2 \times 2$ matrices, ternary matrices $W_a$ and $W_b$ with sizes of $7 \times 4$ can multiply them using 7 multiplications and 36 additions. It is important to note that Strasssen's algorithm requires a hidden layer with 7 units here to compute the exact product matrix that a naïve matrix multiplication algorithm can obtain using 8 multiplications.

Building on top of Strassen's matrix multiplication algorithm, the StrassenNets work (Tschannen et al. (2018)) instead realizes approximate matrix multiplications in DNN layers[1] using fewer hidden layer units compared to the standard Strassen's algorithm required to achieve the exact product matrix. StrassenNets makes this possible by training a SPN-based DNN framework end-to-end to learn the ternary weight matrices from the training data. The learned ternary weight matrices can then approximate the otherwise exact matrix multiplications of the DNN layers with significantly fewer multiplications than Strassen's algorithm. The approximate transforms realized by the SPNs, adapted to the DNN architecture and application data under consideration, can enable precise control over the number of multiplications and additions required per inference, creating an opportunity to tune DNN models to strike an optimal balance between accuracy and computational complexity.

---

[1]A convolutional operation in DNN layers can be reduced to a general matrix multiplication (GEMM). In the context of strassenified matrix multiplications of a network layer, $A$ is associated with the weights or filters of the layer and $B$ is associated with the corresponding activations or feature maps. As a result, after training, $W_a$ and $vec(A)$ can be collapsed into a vector $\hat{a} = W_a vec(A)$, as they are both fixed during inference.

Table 1: Test accuracy along with the number of multiplications, additions, operations and model size for MobileNets-V1 and strassenified MobileNets-V1 (ST-MobileNets) with the width multiplier $0.5$ on ImageNet dataset. $r$ is the hidden layer width of a strassenified convolution layer, $c_{out}$ is the number of output channels of the corresponding convolution layer. A multiply-accumulate operation is abbreviated as MAC.

| Network | Accuracy (%) | Muls | Adds | MACs | Model size | Energy/inference (normalized) | Throughput (normalized) |
|---|---|---|---|---|---|---|---|
| MobileNets (float16) | 65.2 | - | - | 149.49M | 2590.07KB | 1 | 1 |
| MobileNets (TWN (Li & Liu (2016))) | 55.54 | - | 149.49 | - | 323.75KB | 0.2 | 2 |
| ST-MobileNets ($r = 0.5c_{out}$) | 48.92 | 0.77M | 158.54M | 8.69M | 522.33KB | 0.27 | 1.69 |
| ST-MobileNets ($r = 0.75c_{out}$) | 56.95 | 1.16M | 236.16M | 8.69M | 631.76KB | 0.37 | 1.17 |
| ST-MobileNets ($r = c_{out}$) | 61.8 | 1.55M | 313.78M | 8.69M | 741.19KB | 0.48 | 0.9 |
| ST-MobileNets ($r = 2c_{out}$) | 65.14 | 3.11M | 624.27M | 8.69M | 1178.92KB | 0.9 | 0.46 |

The success of StrassenNets in achieving significant compression for $3 \times 3$ convolutions (Tschannen et al. (2018)) and increasing visibility of DS convolutions in resource-constrained networks inspired us to apply StrassenNets over the already compute-efficient MobileNets architecture to reduce its computational costs and model size even further. Further compression of DS layers will not only enable more energy-efficient networks leading to longer lasting batteries, but also will open up the opportunities for more complex use-cases to fit in the limited memory budget of emergent DNN hardware accelerators. Among the various MobileNets architectures (Howard et al. (2017); Sandler et al. (2018); Howard et al. (2019)), in this work we extensively study the quantization of MobileNets-V1 (Howard et al. (2017)). MobileNets-V1 stacks one 3x3 and 13 DS convolutional layers. A DS convolution first convolves each channel in the input feature map with a separate 2-D filter (depthwise convolution) and then uses 1x1 pointwise convolutions to combine the outputs in the depth dimension.

### 2.2.1 STRASSENNETS FOR MOBILENETS

We observe that although strassenifying MobileNets reduces multiplications significantly as expected, it increases additions considerably in order to achieve an accuracy comparable to that of the state-of-the-art MobileNets with 16-bit floating-point weights. Table 1 captures our observation. The strassenified network with the $r = 2c_{out}$ configuration achieves a comparable accuracy to that of the full-precision MobileNets while reducing multiplications by $97.91\%$ but increasing additions by $317.59\%$ (149.49M MACs of MobileNets vs. 3.11M multiplications and 624.27M additions of ST-MobileNets with $r = 2c_{out}$). This in turn offers modest savings in energy required per inference but causes significant degradation in throughput (see Section 4 for details). As shown in Table 1, a number of potential values for the hidden layer width ($r$) were explored. Using fewer hidden units e.g. $r = c_{out}$ than this incurs a siginificant accuracy loss of $3.4\%$.

### 2.2.2 COMPUTE INEFFICIENCY OF STRASSENNETS FOR MOBILENETS

It is important to note here that although the number of additions does increase marginally with strassenifying standard $3 \times 3$ or $5 \times 5$ convolutional layers (Tschannen et al. (2018)), that trend does not hold true with strassenifying MobileNets dominated with DS layers. This stems from the fact that $1 \times 1$ pointwise convolutions dominate the compute bandwidth of a neural network with DS layers (Howard et al. (2017)) and strassenifying a $1 \times 1$ pointwise convolution requires executing two equal-sized (for $r = c_{out}$) $1 \times 1$ convolution operations (with ternary weight filters) in place of the standard $1 \times 1$ convolution, as shown in Figure 2(a). This results in a significant increase (2 : 1 or $100\%$) in additions in comparison to the execution of the standard $1 \times 1$ convolution. In contrast to that, as Figure 2(a) illustrates, a $3 \times 3$ strassenified convolution with $r = c_{out}$ instead requires executing a $3 \times 3$ convolution and a $1 \times 1$ convolution with ternary weight filters, causing a marginal increase (10 : 9 or $11.1\%$) in additions compared to the execution of the standard $3 \times 3$ convolution.

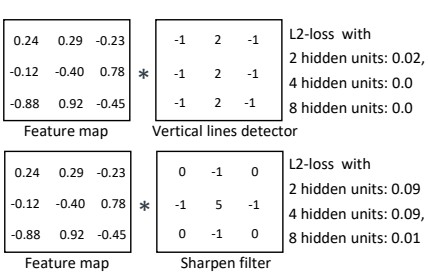

(a) Variance in the sensitivity of filters to quantization.

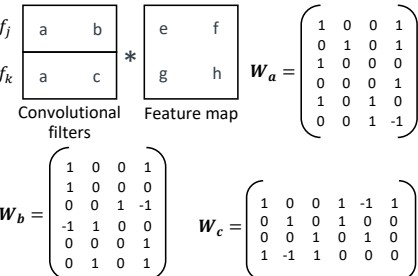

(b) Ease of ternary quantization for a filter bank with common values.

Figure 1: Understanding the sensitivity of individual and group of filters to ternary quantization.

This overhead of addition operations with strassenified DS convolutions increases in proportion to the width of the strassenified hidden layers, i.e. to the size of the ternary convolution operations, as observed in Table 1. As a result, a strassenified DS convolution layer may incur enough overhead to offset the benefit of strassenifying a DS convolution layer.

While Tschannen et al. (2018) demonstrates better trade-offs requiring a modest (29.63%) increase in additions when strassenifying ResNet-18 architecture dominated with $3 \times 3$ convolutions, this does not continue once StrassenNets is applied over MobileNets. This also indicates that the DS convolutions, owing to efficiency in number of parameters than $3 \times 3$ convolutions, are more prone to quantization error and this manifests when StrassenNets is applied. Considering the fact that MAC operations typically consume about five times more energy than addition operations for 16-bit floating-point values (Horowitz (2014); Andri et al. (2018)) (see Section 4 for details), an about 317.59% increase in additions in place of about 98% saving on multiplications will result in diminishing or no returns in terms of energy savings and runtime speedups even on specialized hardware dominated with adders. This increase in computational costs associated with strassenified DS convolutions in conjunction with the high accuracy and low latency requirements of mobile applications call for a model architecture exploration that can leverage the compute efficiency of DS layers and model size reduction of strassenified convolutions while maintaining acceptable or no increase in additions.

The accuracy drop using a strassenified MobileNets with the $r = c_{out}$ configuration essentially indicates that each layer perhaps introduces a certain amount of quantization error owing to lower hidden width and that error accrues over multiple quantized layers. On the other hand, although a strassenified MobileNets with $r = 2c_{out}$ recovers the accuracy loss of the $r = c_{out}$ configuration, it makes a strong assumption that all filters require wider strassenified hidden layers to quantize to ternary bits to preserve the representational power of the baseline full-precision network. While this might be true for some of the convolutional filters, not all filters need to be quantized using the $r = 2c_{out}$ configuration. This observation stems from the following two reasons:

**(a) Different sensitivity of individual filters to StrassenNets.** Different convolutional filters tend to extract different type of features, ranging from simple features (e.g. edge detection) to more complicated higher-level (e.g. facial shapes) or object specific features. As a result, different filters may respond differently to ternary quantization. That basically means there are filters that are easy to quantize to ternary values using narrower hidden layers while still ensuring low L2 reconstruction error in output feature maps.On the other hand, there are weight filters that require wider strassenified hidden layers to ensure a low or modest L2 loss.

Given a feature map, Figure 1(a) presents a scenario where a strassenified vertical lines detector with fewer hidden layer units can closely approximate the output map (with low L2 reconstruction loss) produced otherwise using its full-precision counterpart. However a convolutional filter that sharpen images requires a wider hidden layer to ensure a low L2 loss (see Appendix C.1 for more details). Note that we only consider 2D filters for illustration purpose, whereas this difference in complexity should exist in 3D filters common to CNNs.

**(b) Different sensitivity of group of filters to StrassenNets.** Furthermore, there exists groups of convolutional filters at each layer that either tend to extract fairly similar features with slightly different orientations (e.g. two filters attempting to detect edges rotated by few degrees) or have other numerical-structural similarities. As a result, when these groups of convolutional filters are quantized to ternary values using StrassenNets, they may share many hidden layer elements. These groups of convolutional filters with similar value structure in turn are more amenable to quantization using fewer hidden layer units than filters with no common value structure. Given a constrained hidden layer budget for StrassenNets (e.g. $r = c_{out}$), these groups of convolutional filters may together respond well to ternary quantization while other dissimilar filters struggle to be strassenified alongside them with low quantization error, due to the restricted hidden layer bandwidth.

Figure 1(b) illustrates a case when two filters $f_j$ and $f_k$, having some common value structure, can learn to perform exact convolution with a $2 \times 2$ feature map using only 6 multiplications instead of the 7 required otherwise for unique filters lacking common value structure. A set of ternary weight matrices with fewer hidden units implementing an exact convolution in this case is shown in Figure 1(b) (see Appendix A for more details).

Motivated by these observations, we propose a novel quantization method – one that will only quantize *easy-to-quantize* weight filters of a network layer to ternary values (to restrict the increase in additions) while also preserving the representational ability of the overall network by relying on few full-precision *difficult-to-quantize* weight filters. This layer-wise hybrid filter bank strategy exploits a full-precision network's strength as a highly-accurate classifier and couples that with Strassen-Nets to achieve significant reduction in model size and number of multiplications. This quantization technique essentially maintains a good balance between overall computational costs and predictive performance of the overall network.

## 3 PER-LAYER HYBRID FILTER BANKS

We propose a quantization method that can quantize a substantial fraction of convolutional filters to ternary values at each layer while relying on few remaining full-precision filters to preserve the representational power of the original full-precision network. *As easy-to-quantize filters are quantized only using StrassenNets leaving the difficult-to-quantize filters in full-precision values, this should in turn require narrow hidden layers for quantizing them resulting in an overall reduction in computations (additions along with MAC operations) and memory footprint while ensuring no loss in accuracy.* This is in sharp contrast to quantizing all the filters of each layer using wide hidden layers to preserve the representational power of MobileNets which led to significant increase in additions as we have seen in Section 2.2.1.

**Architecture.** The proposed quantization method convolves the same input feature map with full precision weight filters and ternary weight filters in parallel, concatenating the feature maps from each convolutions into an unified feature map. This concatenated feature map is fed as input to the next network layer. At each layer, the combination of the two convolutions from full-precision and ternary filters ensures that they combine to form a output feature map of identical shape as in the baseline full-precision network. For instance, given an input feature map with $c_{in}$ channels, the quantization technique applies traditional convolution with $k$ full-precision weight filters $W_{fp}$ of shape $c_{in} \times w_k \times h_k$ and strassen convolution with $c_{out} - k$ ternary weight filters $W_t$ to produce a feature map of total $c_{out}$ channels for a layer. Here $c_{out}$ is the number of channels in the output volume of the corresponding convolution layer in the baseline full-precision network, and $w_k, h_k$ are the kernel size. For the sake of simplicity, bias term is not included in this discussion. The fraction of channels generated in an output feature map from the full-precision weight filters, $\alpha$ (or in others words the channels generated from the ternary weight filters, $1 - \alpha$) is a hyperparameter in our quantization technique and it decides the representational power and computational costs of MobileNets with hybrid filter banks.

Figure 2(b) shows the organization of the hybrid filter bank for a MobileNets layer. Each of the convolutional layers of MobileNets, including the $3 \times 3$ layer and the $1 \times 1$ pointwise convolutions of the following 13 depthwise-separable layers, are quantized using hybrid filter banks, where $\alpha\%$ of output channels at each layer is generated using full-precision weight filters and the remaining output channels using ternary weight filters. The depthwise convolutions of the depthwise-separable layers are not quantized using either StrassenNets or our hybrid filter banks. This is primarily due

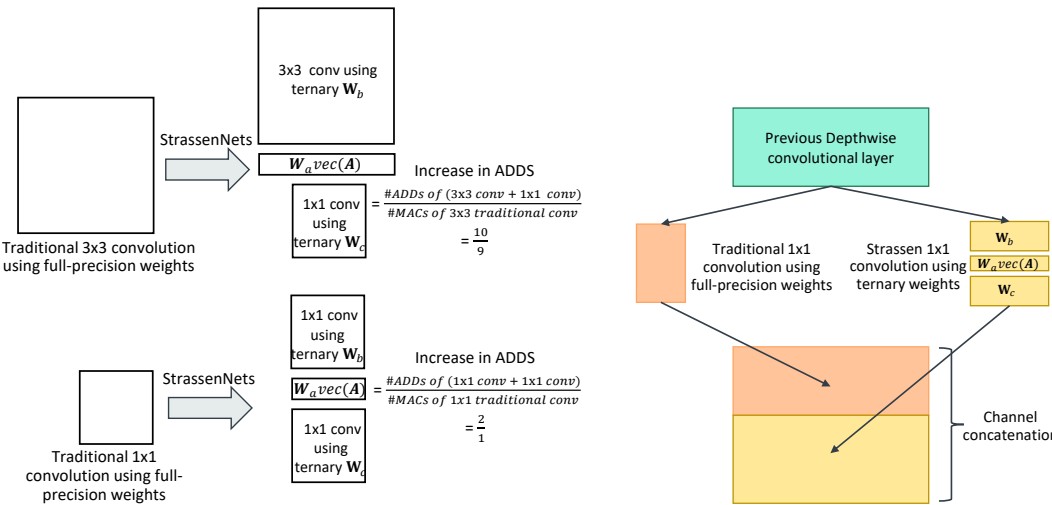

(a) Application of StrassenNets to 3 x 3 and 1 x 1 convolution. The cost of elementwise multiplication with intermediate $W_a vec(A)$ is comparably negligible and hence is ignored in estimating the increase in additions.

(b) A MobileNets pointwise layer with hybrid filter bank.

Figure 2: MobileNets with hybrid filter banks.

to the following reasons: (a) they do not dominate the compute bandwidth of MobileNets (Howard et al. (2017)), (b) as per our observations, quantizing those to ternary values hurt the accuracy significantly without offering any significant savings in either model size or computational costs. The strassenified convolutions portion of hybrid filter banks at each layer are quantized using a number of $r$ values, where $r$ is the hidden layer width of a strassenified convolution layer. *The $r << 2c_{out}$ configuration in conjunction with an optimal non-zero $\alpha$ should offer substantial savings in model size and addition operations without compromising accuracy in comparison to a fully strassenified MobileNets architecture with $r = 2c_{out}$ configuration.* The presented quantization technique can also be applied to the fully-connected layer parameters, however, we only focus on convolution layers in this work. We compress the last fully-connected layer of MobileNets uniformly using StrassenNets. The per-layer hybrid filter banks proposed here is inspired by the Inception module from the GoogLeNet architecture (Szegedy et al. (2015)) (see Appendix B for more details).

**End-to-end training.** The full-precision filters along with the strassenified weight filters for each layer are trained jointly so as to maximize accuracy. A gradient-descent (GD) based training algorithm is used to train the network with hybrid filter banks end-to-end. Before the training begins, depending on the value of $\alpha$, the top $\alpha * c_{out}$ channels of a feature map are configured to generate from full-precision traditional convolutions, and the remaining $1 - \alpha * c_{out}$ channels are forced to generate from ternary strassenified convolutions. Note that the order of the channels generated in the output feature volume by either full-precision filters or ternary filters is not important, as the output feature map comprising all the channels generated forms the input of the subsequent layer and the weights in the subsequent layer can adjust to accommodate that. During the end-to-end training process, the organization of hybrid filter banks tend to influence the difficult-to-quantize filters (that require full-precision filters to extract features) to be trained using full-precision values, and the filters that are less susceptible to ternary quantizationto be trained using ternary values from strassenified convolutions. Furthermore, in order to recover any accuracy loss of the hybrid network compressed with strassenified matrix computations, knowledge distillation (KD) is exploited during training, as described in( Tschannen et al. (2018)). Using KD, an uncompressed teacher network can transfer its prediction ability to a compressed student network by navigating its training. We use the uncompressed hybrid network as the teacher network and the compressed strassenified network as the student network here.

## 4 EXPERIMENTS AND RESULTS

**Datasets and experimental setup.** We evaluate the MobileNets-V1 architecture comprising proposed per-layer hybrid filter banks (Hybrid MobileNets) on the ImageNet (ILSVRC2012)

dataset (Deng et al. (2009)) and compare it against the state-of-the-art MobileNets (Howard et al. (2017)) with 16-bit floating-point weights. The baseline and other network architectures presented here use a width multiplier of $0.5^2$ to reduce training costs with limited GPU resources. We use the MXNet framework (Chen et al. (2015)) based GluonCV toolkit[3] to train the networks. This is primarily attributed to the better top-1 accuracy (65.2%) of MobileNets-V1 (width multipler of 0.5) achieved by the GluonCV toolkit[4] when compared to the top-1 accuracy of 63.3% observed by the corresponding publicly available model in the Tensorflow framework (Abadi et al. (2016)). In this work, the baseline MobileNets and the full-precision filters of the hybrid filter banks use 16-bit floating-point weights. We quantize the activations of the baseline and proposed architectures to 16-bit floating-point values. A 8-bit representation of weights and activations should not alter the conclusions made in this work. At the time of writing this paper, GluonCV toolkit does not support training with 8-bit weights and activations.

**Hybrid MobileNets architecture training.**    We use the Nesterov accelerated gradient (NAG) optimization algorithm and follow the other training hyperparameters described in the GluonCV framework for training the baseline full-precision MobileNets, strassenified MobileNets and our proposed Hybrid MobileNets. We begin by training the Hybrid MobileNets with full-precision strassen matrices ($W_a$, $W_b$, and $W_c$) for 200 epochs. With a mini-batch size per GPU of 128 on a 4 GPU system, the learning rate is initially chosen as 0.2, and later gradually reduced to zero following a cosine decay function as used in the GluonCV framework for training the baseline full-precision MobileNets (see Appendix C.2 for more details).

We then activate quantization for these strassen matrices and the training continues for another 75 epochs with initial learning rate of 0.02 and progressively smaller learning rates. Quantization converts a full-precision strassen matrix to a ternary-valued matrix along with a scaling factor (e.g., $W_b$ = scaling factor * $W_b^t$). To evaluate our hypothesis that some full-precision filters are changing significantly to recover features lost due to quantization, we measured the L2 distance between their pre- and post-quantization weight vectors. We found the L2 distances fit a normal distribution: most filters experience low-to-moderate changes to their weight vectors while a few exceptional filters saw very significant movement. This supports our claim that the full-precision filters are preserving the overall representational power of the network.

Finally, we fix the strassen matrices of the hybrid filter banks to their learned ternary values and continue training for another 25 epochs to ensure that the scaling factors associated with these matrices can be absorbed by full-precision $vec(A)$ portion of strassenified matrix multiplication.

**Energy and throughput modeling for hybrid filter banks.**    The proposed per-layer hybrid filter banks for MobileNets can be executed by existing DNN hardware accelerators, such as DaDianNao (Chen et al. (2014)) and TPU (Jouppi et al. (2017)) consisting of only MAC units. However, in order to achieve an energy- and runtime- efficient execution of hybrid filter banks dominated with additions, we propose a custom hardware accelerator, where a fraction of MAC units are replaced by low-cost adders within the same silicon area. A 16-bit floating-point MAC unit takes about twice the area of a 16-bit floating-point adder (Lutz (2019)). Given a fixed silicon area and a model configuration for Hybrid MobileNets, the ratio of MAC units to adders in the proposed hardware accelerator is decided in such a way that the maximum possible throughput can be achieved for the configuration. In order to estimate the energy required per inference of baseline and proposed models, we use the energy consumption numbers of 16-bit floating-point adder and MAC unit mentioned in (Horowitz (2014)).

**Hybrid MobileNets architecture evaluation.**    One of the main focus of our evaluation is the study of how $\alpha$ impacts on the performance of our models. This parameter, that can be independently set for each convolutional layer in the network, is directly proportional to the number of learnable parameters in a given layer. In this work, we use identical value of $\alpha$ for all the layers of Hybrid MobileNets. We believe use of different values for different layers may result in better cost accuracy trade-offs. We leave this exploration for future work. Ideally small values of $\alpha$ and $r$ are desired to

---

[2]Using a width multiplier of 0.5 halves the number of channels used in each layer of the original MobileNets architecture (Howard et al. (2017)).

[3] GluonCV: a Deep Learning Toolkit for Computer Vision, https://gluon-cv.mxnet.io/index.html

[4]https://gluon-cv.mxnet.io/model_zoo/classification.html#mobilenet

[5]As this configuration is likely to observe an accuracy of $\leq 63.47$, we did not collect the accuracy result for this configuration.

Table 2: Top-1 accuracy along with the computational costs, model size, and energy per inference for baseline MobileNets-V1, ST-MobileNets, and Hybrid MobileNets on ImageNet dataset. $\alpha$ is the fraction of channels generated by the full-precision weight filters at each layer , $c_{out}$ is the number of remaining channels generated by the ternary strassen filters at the corresponding convolutional layer, $r$ is the hidden layer width of the strassenified convolutions. The last column shows the throughput of proposed models on an area-equivalent hardware accelerator comprising both MAC and adder units when compared to the throughput of baseline MobileNets with 16-bit floating-point weights on a MAC-only accelerator.

| Network | Alpha ($\alpha$) | $r$ | Acc. (%) | Muls, Adds | MACs | Model size | Energy/inference (normalized) | Throughput (normalized) |
|---|---|---|---|---|---|---|---|---|
| MobileNets (float16) | - | - | 65.2 | - | 149.49M | 2590.07KB | 1 | 1 |
| ST-MobileNets | 0 | $2c_{out}$ | 65.14 | 3.11M, 624.27M | 8.69M | 1178.92KB | 0.9 | 0.46 |
| MobileNets (Hybrid filter banks) | 0.25 | $c_{out}$[5] | - | 1.16M, 204.63M | 43.76M | 1004.67KB | 0.56 | 1.02 |
| | | $1.33c_{out}$ | 63.47 | 1.55M, 270.95M | 43.76M | 1097.07KB | 0.65 | 0.83 |
| | | $2c_{out}$ | 65.2 | 2.33M, 405.59M | 43.76M | 1284.65KB | 0.84 | 0.6 |
| MobileNets (Hybrid filter banks) | 0.375 | $c_{out}$ | 64.13 | 0.97M, 157.84M | 61.3M | 1131.43KB | 0.62 | 1.06 |
| | | $1.6c_{out}$ | 64.17 | 1.55M, 250.34M | 61.3M | 1260.44KB | 0.74 | 0.8 |
| | | $2c_{out}$ | 65.2 | 1.94M, 312.01M | 61.3M | 1346.45KB | 0.83 | 0.68 |
| MobileNets (Hybrid filter banks) | 0.5 | $c_{out}$ | 64.69 | **1.28M, 142.37M** | **78.83M** | **1267.13KB** | **0.72** | **1** |
| | | $2c_{out}$ | 65.17 | 1.55M, 228.68M | 78.83M | 1327.88KB | 0.83 | 0.77 |

achieve significant reduction in MAC along with addition operations while preserving the baseline accuracy.

We search the model hyperparameters space systematically to develop Hybrid MobileNets. Table 2 captures the top-1 accuracy of the Hybrid MobileNets for various configurations of $\alpha$ and hidden layer width $r$, along with their impact on computational costs, model size, energy required per inference, and throughput and and compares that against baseline full-precision MobileNets, and ST-MobileNets. As shown in Table 2, the ST-MobileNets and various configurations of Hybrid MobileNets offer comparable reduction (about $50\%$) in model size over the baseline full-precision Mobilenets. While the $r = 2c_{out}$ configurations for different values of $\alpha$ (0.25, 0.375, and 0.5) can preserve the baseline top-1 accuracy of $65.2\%$ and offer modest savings in energy required per inference, that comes at the cost of large increase in additions. This in turn causes significant degradation in throughput on the proposed hardware accelerator when compared to the throughput of the baseline full-precision MobileNets on an existing DNN accelerator consisting of only MAC units. On the other end, the $c_{out} \leq r < 2c_{out}$ configurations with the $\alpha$ of 0.25 and 0.375 incur modest to significant drop in top-1 accuracy possibly owing to lack of enough full-precision weights filters at each hybrid filter bank to preserve the representational ability of the overall network. The $r < c_{out}$ configurations for different values of $\alpha$ leads to large drop in prediction accuracy and hence is not shown in Table 2.

The Hybrid MobileNets with the $\alpha = 0.5$ and $r = c_{out}$ configuration strikes an optimal balance between accuracy, computational costs, energy, and throughput. It achieves comparable accuracy to that of the baseline MobileNets, strassenified and Hybrid MobileNets with the $r = 2c_{out}$ configuration while reducing the number of MACs, and multiplications by $47.26\%$, and $46.4\%$ respectively and requiring a modest ($45.51\%$) increase in additions over the baseline MobileNets architecture. Of particular note is that it reduces the number of additions to about $142.37M$ when compared to $624.27M$ additions of ST-MobileNets described in Section 2. The significant reduction in MAC operations and modest increase in additions over the baseline full-precision MobileNets in turn translates into $27.98\%$ savings in energy required per inference while ensuring no degradation in throughput in comparison to the execution of baseline MobileNets on a MAC-only hardware accelerator. This reduction in additions is primarily attributed to strassenifying easy-to-quantize filters using fewer hidden units ($r = c_{out}$) while relying on full-precision filters to generate $50\%$ channels at each layer and preserve the representational ability of the overall MobileNets architecture. Owing to the substantial presence of ternary weights matrices, the Hybrid MobileNets with the $\alpha = 0.5$ and

$r = c_{out}$ configuration reduces the model size to $1267.13$KB when compared to $2590.07$KB of the baseline MobileNets network thus enabling a $51.07\%$ savings in model size. The use of knowledge distillation in training the ST-MobileNets and Hybrid MobileNets does not result in any tangible change in accuracy.

*In summary, the Hybrid MobileNets reduces model size by* $51.07\%$ *and energy required per inference by* $27.98\%$ *while incurring a negligible loss in accuracy and no degradation in throughput when compared to the baseline full-precision MobileNets.* It is important to note that because of the large savings in model size, our Hybrid MobileNets will have significantly fewer accesses to the energy/power-hungry DRAM. This in conjunction with skipping ineffectual computations of zero-valued weights in our proposed hardware accelerator (as exploited by (Zhang et al. (2016))), owing to about $40 - 50\%$ of sparsity in the ternary weight matrices of strassenified layers as we observe, will improve the energy savings and run-time performance even further. Our current energy and throughput modeling does not take this into account. We leave this exploration for future work.

## 5 RELATED WORK

**Weight pruning.** Sparsifying filters and pruning channels are widely used methods to make neural networks more resource-efficient. Unstructured filter sparsity inducing techniques either observe poor hardware characteristics or incur modest to significant drop in model accuracy for MobileNets (Zhu & Gupta (2017)). Recent work on channel pruning (He et al. (2018)) demonstrates negligible drop in accuracy for MobileNets while achieving significant reduction in computational costs. As different channel pruning (He et al. (2018); Zhuang et al. (2018); He et al. (2017)) and filter pruning techniques (Han et al. (2015); Narang et al. (2017); Zhu & Gupta (2017); Guo et al. (2016); Aghasi et al. (2017); Wen et al. (2016); Luo et al. (2017); Yang et al. (2018); Gordon et al. (2018)) are orthogonal to our compression scheme, they can be used in conjunction with Hybrid MobileNets to further reduce model size and computational complexity.

**Network quantization.** Recent works on binary/ternary quantization either do not demonstrate their potential to quantize MobileNets on ImageNet dataset (Yang et al. (2019); Zhuang et al. (2019); Zhu et al. (2019); Sun et al. (2019); Zhang et al. (2018a); Guo et al. (2017)) or incur modest to significant drop in accuracy while quantizing MobileNets with 4-6-bit weights (Wang et al. (2019); Liu & Mattina (2019); Louizos et al. (2019)) (see Appendix D for more details). The hybrid filter banks successfully quantizes a significant fraction of weight filters of MobileNets to ternary values while achieving comparable accuracy to that of baseline full-precision model on ImageNet. Nevertheless, the hybrid filter banks can benefit further by adopting these prior proposals.

**Tensor decomposition.** Besides pruning and quantization, tensor decomposition techniques (Jaderberg et al. (2014); Tai et al. (2015); Wen et al. (2017)) exploit parameter redundancy to obtain low-rank approximations of weight matrices without compromising model accuracy. Full-precision weights filters and Strassen matrices of our hybrid filter banks can adopt these prior proposals to further reduce model size and computational complexity.

**Compact network architectures.** While we show promising results for MobileNets-V1 here, the benefits of hybrid filter banks should scale when extended to other popular resource-efficient architectures dominated with either DS convolutions, such as MobileNets-V2 (Sandler et al. (2018)), ShuffleNet (Zhang et al. (2018b)), and Xception (Chollet (2017)) or standard $3 \times 3$ convolutions.

## 6 CONCLUSION AND FUTURE WORK

In this work, we propose per-layer hybrid filter banks for MobileNets capable of quantizing its weights to ternary values while exhibiting start-of-the-art accuracy on a large-scale dataset and requiring a fraction of the model size and considerably lower energy per inference pass. We use 16-bit floating-point format to represent the intermediate activations and traditional weight filters of hybrid filter banks in this work. In future, we plan to explore the impact of quantizing them to 8-bit or less. In addition, it will be interesting to see how channel pruning (He et al. (2018); Zhuang et al. (2018)) assists in reducing the computational complexity of strassenified MobileNets.

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

# A    FAST MATRIX MULTIPLICATIONS VIA STRASSEN'S ALGORITHM

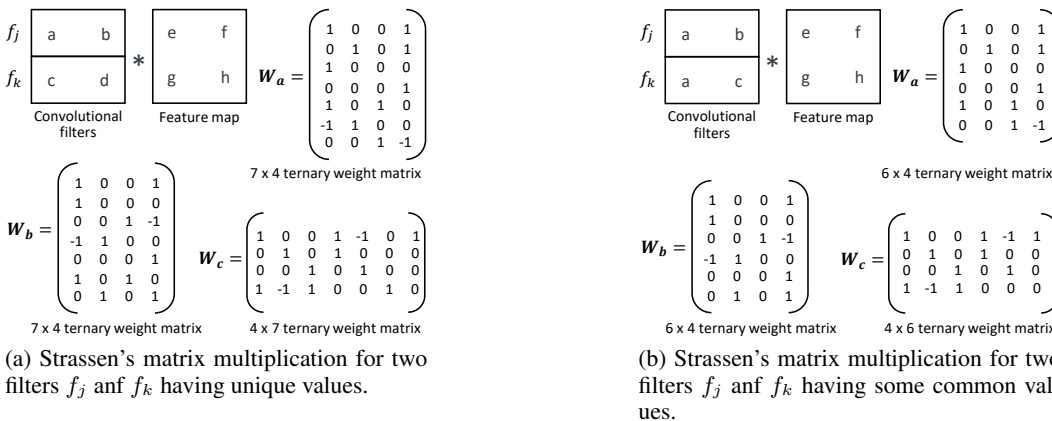

(a) Strassen's matrix multiplication for two filters $f_j$ anf $f_k$ having unique values.

(b) Strassen's matrix multiplication for two filters $f_j$ anf $f_k$ having some common values.

Figure 3: Understanding the sensitivity of Strassen's algorithm to filter values.

Strasssen's algorithm can multiply $2 \times 2$ matrices using only 7 multiplications instead of 8 required otherwise by a naïve matrix multiplication algorithm. Figure 3(a) specifies a set of weight matrices that can perform exact convolution of the $2 \times 2$ filter bank comprising $f_j$ and $f_k$ with the feature map using 7 multiplications. Note that the two filters $f_j$ and $f_k$ do not have any common values. However, owing to the presence of common value of $a$ between $f_j$ and $f_k$ filters in Figure 3(b), Strassen's algorithm now can compute the exact product matrix using only 6 multiplications instead of 7 required otherwise in Figure 3(a). A set of ternary weight matrices implementing an exact convolution in this case is shown in Figure 3(b).

# B    RELATION OF PER-LAYER HYBRID FILTER BANKS TO GOOGLENET ARCHITECTURE.

The per-layer hybrid filter banks proposed here is inspired by the Inception module from the GoogLeNet architecture (Szegedy et al. (2015)). In a traditional convolutional network, each layer extracts information from the previous layer in order to transform the input data into a more useful representation. However, salient features of an input volume can have extremely large variation in size. Because of this variation in the size of the required information, choosing the right kernel size

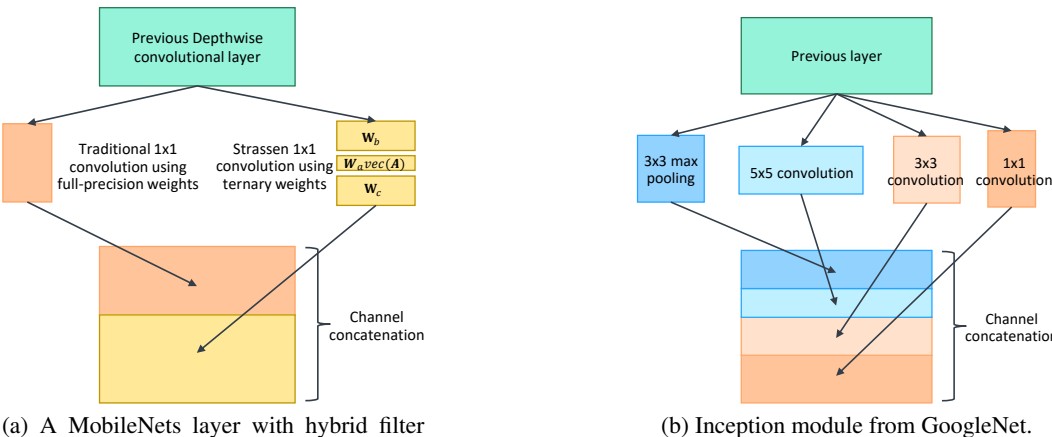

(a) A MobileNets layer with hybrid filter bank.

(b) Inception module from GoogLeNet.

Figure 4: MobileNets with hybrid filter banks and its relation to Inception module from GoogLeNet.

Table 3: Hyperparameters for training Hybrid MobileNets

| Training phase | Hyperparameters |
|---|---|
| Train using full-precision strassen matrices | Batch size per GPU: 128
Number of GPUs used: 4
Optimizer: Nesterov accelerated gradient (NAG)
(Momentum: 0.9, Weight decay: 0.0001)
Number of epochs: 200
Weight initialization: Xavier
Initial, final learning rate: 0.2, 0.0
Learning rate schedule: cosine decay
Number of warmup epochs: 5
Starting warmup learning rate: 0.0
Size of the input image: 224 x 224 x 3 |
| Activate quantization for strassen matrices | Batch size per GPU: 128
Number of GPUs used: 4
Optimizer: Nesterov accelerated gradient (NAG)
(Momentum: 0.9, Weight decay: 0.0001)
Number of epochs: 75
Initial, final learning rate: 0.02, 0.0
Learning rate schedule: cosine decay |
| Freeze strassen matrices to ternary values | Batch size per GPU: 128
Number of GPUs used: 4
Optimizer: Nesterov accelerated gradient (NAG)
(Momentum: 0.9, Weight decay: 0.0001)
Number of epochs: 25
Initial, final learning rate: 0.002, 0.0
Learning rate schedule: cosine decay |

for the convolution operation becomes difficult. The Inception module addresses this by allowing the GoogLeNet architecture to use convolutional filters of different sizes – a small sized $(1 \times 1)$ filter convolution, a medium sized $(3 \times 3)$ filter convolution, and a large sized $(5 \times 5)$ filter convolution at each layer and let the network decide for itself the appropriate convolutional filter to capture the necessary features. The $1 \times 1$ or $3 \times 3$ convolutions cover a small receptive field of the input and can capture fine grain details and features in the input volume, whereas the $5 \times 5$ filters are able to cover a large receptive field, and thus can capture spread out features of higher abstraction.

The proposed quantization technique instead allows the different convolutional filters at each layer to decide its acceptable precision level (among full-precision and ternary quantization in this work) to derive the best possible representational power of the network. Figure 4(a) shows a hybrid filter bank and Figure 4(b) compares that to the Inception module from the GoogleNet architecture.

## C   TRAINING DETAILS

### C.1   SENSITIVITY OF CONVOLUTIONAL FILTERS TO STRASSENNETS

We generate a training set containing 100k pairs $(A_i, B_i)$ with values i.i.d. uniform on $[-1, 1]$ in $A_i$, and values of a given convolutional filter in $B_i$. The SPN is then trained using different number of hidden units. We begin training with full-precision weights (initialized i.i.d. uniform on $[-1, 1]$) for one epoch with SGD (learning rate $0.1$, momentum $0.9$, mini-batch size $4$), activate quantization, and train for few epochs with initial learning rate of $0.01$ and progressively smaller learning rates. Once the training converges after activation of the quantization, we collect the L2-loss.

## C.2 Hyperparameters Settings for Training Hybrid MobileNets

The training images from ImageNet are preprocessed by using mean and standard deviation. These images are resized such that the shorter side has length of $256$ and are then randomly cropped to $224 \times 224$ pixels. Random horizontal flips are applied for data augmentation. The center $224 \times 224$ crop of the images are used for evaluation.

Table 3 shows the hyperparameters values used for training Hybrid MobileNets. Similar hyperparameters values are used for training baseline full-precision MobileNets and ST-MobileNets also. The learning rate scheduling involves a 'warm up' period in which the learning rate is annealed from zero to $0.2$ over the first $5$ epochs, after which it is gradually reduced following a cosine decay function.

# D Comparison against Prior Works

Table 4: Top-1 and top-5 accuracy (%) of Mobilenet (full resolution and multiplier of $0.5$) on Imagenet for different number of bits per weight and activation.

| Method | #bits per weight/activation | Top-1 Acc. (%) | Top-5 Acc. (%) |
|---|---|---|---|
| Baseline MobileNets[6] | 32/32 | 65.53 | 86.48 |
| Baseline MobileNets[7] | 16/16 | 65.2 | 86.34 |
| ST-MobileNets ($r = 0.5c_{out}$) | 2/16 | 48.92 | 73.68 |
| ST-MobileNets ($r = 0.75c_{out}$) | 2/16 | 56.95 | 80.25 |
| ST-MobileNets ($r = c_{out}$) | 2/16 | 61.8 | 83.97 |
| ST-MobileNets ($r = 2c_{out}$) | 2/16 | 65.14 | 86.26 |
| Hybrid MobileNets ($\alpha = 0.25, r = 1.33c_{out}$) | 2,16/16 | 63.47 | 85.1 |
| Hybrid MobileNets ($\alpha = 0.25, r = 2c_{out}$) | 2,16/16 | 65.2 | 86.05 |
| Hybrid MobileNets ($\alpha = 0.375, r = c_{out}$) | 2,16/16 | 64.13 | 85.4 |
| Hybrid MobileNets ($\alpha = 0.375, r = 1.6c_{out}$) | 2,16/16 | 64.17 | 85.38 |
| Hybrid MobileNets ($\alpha = 0.375, r = 2c_{out}$) | 2,16/16 | 65.2 | 86.05 |
| Hybrid MobileNets ($\alpha = 0.5, r = c_{out}$) | 2,16/16 | 64.69 | 85.66 |
| Hybrid MobileNets ($\alpha = 0.5, r = 2c_{out}$) | 2,16/16 | 65.17 | 85.98 |
| Baseline MobileNets[8] | 32/32 | 63.3 | 84.9 |
| Baseline MobileNets[9] | 8/8 | 62.2 | - |
| Alpha-blending (Liu & Mattina (2019)) | 8/8 | 63 | - |
| Alpha-blending (Liu & Mattina (2019)) | 4/8 | 58.4 | - |
| HAQ (Wang et al. (2019))[10] | | | |
| RQ (Louizos et al. (2019))[11] | | | |

---

[6]https://gluon-cv.mxnet.io/model_zoo/classification.html#mobilenet

[7]https://gluon-cv.mxnet.io/model_zoo/classification.html#mobilenet

[8]https://github.com/tensorflow/models/blob/master/research/slim/nets/mobilenet_v1.md

[9]https://github.com/tensorflow/tensorflow/tree/r1.14/tensorflow/contrib/quantize

[10]HAQ only shows accuracy results for the width multiplier of 1.

[11]RQ only shows accuracy results for the width multiplier of 1.

