# OpenReview forum: "Ternary MobileNets via Per-Layer Hybrid Filter Banks"
_ICLR.cc/2020/Conference — Reject_

### Official Review · AnonReviewer2 · 2019-10-22
**Official Blind Review #2**

**Rating:** 3

**Review:**

The authors focus on quantizing the MobileNets architecture to ternary values, resulting in less space and compute.

The space of making neural networks more energy efficient is vital towards their deployment in the real world.

I think the authors over-state their claims of no loss in accuracy, in Table 2 we see a clear loss in accuracy from MobileNets to MobileNets + Hybrid Filter Banks.

I think this research is quite incremental over MobileNets and is unlikely to spur further research strains. I think a better venue for this research may be a more systems-focused conference or journal.

There is a significant amount of compute and training complexity required to reduce the model size, e.g. versus model pruning or tensor decomposition. It seems this research would be incredibly difficult to reproduce.

**Experience Assessment:**

I do not know much about this area.

**Review Assessment: Checking Correctness Of Derivations And Theory:**

I did not assess the derivations or theory.

**Review Assessment: Checking Correctness Of Experiments:**

I assessed the sensibility of the experiments.

**Review Assessment: Thoroughness In Paper Reading:**

I made a quick assessment of this paper.

---

> ### Author Response · Authors · 2019-11-15
> **Response to Review #2**
>
> We thank the reviewer for the thoughtful feedback. Please find the responses inline below.
>
> (1) I think the authors over-state their claims of no loss in accuracy, in Table 2 we see a clear loss in accuracy from MobileNets to MobileNets + Hybrid Filter Banks.
>
> Please note that the results reported for hybrid filter banks are from a single run of our training procedure, we believe the 0.51% accuracy drop can be bridged with more training hyperparameter exploration. We will add the updated results with more hyperparameter exploration in the final version. In any case we will change the text appropriately to reflect the reviewer’s concerns.
>
> (2) I think a better venue for this research may be a more systems-focused conference or journal.
>
> While we agree this paper has a system-focused contribution in terms of proposing a novel hardware accelerator consisting of both MAC and adder units for efficient execution of ternary weight networks, the primary focus throughput the paper is the model compression of MobileNets’ already highly optimized network, the discussion of issues around quantizing MobileNets to ternary weights and subsequent development of a ML solution to address that. Hence, we submitted the work to a ML conference where we feel attendees will find more value and interest in this work.
>
> (3) I think this research is quite incremental over MobileNets.
>
> Please see answer to Q1 of Review #1.  Approximately 2x compression in model size and 28% saving in energy with no degradation in inference throughput or little degradation in accuracy is not an incremental contribution for an already compact and compute-efficient architecture like MobileNets.  This paper provides the new best-in-class result for this task.
>
> (4) I think this research is unlikely to spur further research strains.
>
> We strongly believe that this work will spur further research strains because of the following reasons:
> a)	The research community has put significant effort into quantizing ResNet architecture to ternary values while preserving the accuracy of full-precision model [1][2][3][4][5][6][7]. Considering the fact that MobilNet family of networks (1x1 convolutions) have fueled tremendous progress in the design and organization of resource-efficient architectures in recent years, it is highly likely that the MobileNet-like compute-efficient networks will be the next target for ternary quantization.
> b)	To the best of our knowledge, the hybrid filter banks proposed in this work is a first step towards quantizing the already compute-efficient MobileNets architecture to ternary values with a negligible loss in accuracy on a large-scale dataset, such as ImageNet. The Table 2 in Page 13 of reference [4] paper further shows the evidence of our claim.
>
> References:
> (1)	Kuan Wang et al., “HAQ: Hardware-Aware Automated Quantization with Mixed Precision”, CVPR 2019
> (2)	Ron Banner et al., “Post-training 4-bit quantization of convolution networks for rapid-deployment”, NeurIPS 2019
> (3)	Yoni Choukroun et al., “Low-bit Quantization of Neural Networks for Efficient Inference”, ICCV 2019 workshop
> (4)	Christos Louizos et al., “Relaxed Quantization for Discretized Neural Networks”, ICLR 2019
> (5)	Jiwei Yang et al., “Quantization Networks”, CVPR 2019
> (6)	Ruihao Gong et al., “Differentiable Soft Quantization: Bridging Full-Precision and Low-Bit Neural Networks”, ICCV 2019
> (7)	Sangil Jung et al., “Learning to Quantize Deep Networks by Optimizing Quantization Intervals with Task Loss”, CVPR 2019
>
>
> (5) There is a significant amount of compute and training complexity required to reduce the model size, e.g. versus model pruning or tensor decomposition. It seems this research would be incredibly difficult to reproduce.
>
> (a)	The training of hybrid filter banks requires finding the appropriate division of output channels to be generated from either full-precision or ternary weight filters at each layer to explore the best cost-accuracy tradeoffs. Model pruning and tensor decomposition techniques also must find either the appropriate sparsity factors or the appropriate sizes of the tensor decomposed matrices during training to meet the required cost-accuracy tradeoffs. From our experience hybrid filter banks require similar training complexity to that of the training complexity of model pruning and tensor decomposition techniques.
> (b)	Furthermore, note that the model pruning and the tensor decomposition techniques (few cited in section 5, Related Work) are orthogonal to our compression scheme, they can be used in conjunction with hybrid filter banks to further reduce model size and computational complexity (likely at the cost of accuracy).
> (c)	Hybrid filter banks compression scheme is profitable and generalizable to other network architectures (ResNet dominated with 3x3 convolutional layers) as well (please see answer to Q1 to Review# 3).
> (d)	We are happy to release the training infrastructure for hybrid filter banks to ease reproducibility of our work.

---

### Official Review · AnonReviewer1 · 2019-10-24
**Official Blind Review #1**

**Rating:** 3

**Review:**

This paper proposes a novel quantization method towards the MobileNets architecture, with the consideration of balance between accuracy and computation costs. Specifically, the paper proposes a layer-wise hybrid filter banks which only quantizes a fraction of convolutional filters to ternary values while remaining the rest as full-precision filters. The method is tested empirically on ImageNet dataset.

This paper is generally well written with good clarity. Several concerns are as follows:

1. This paper is incremental in nature, with a natural generalization of (Tschannen et al.(2018)). But it is still an interesting contribution. For this kind of paper, I would like to see a more complete set of empirical results. However, The experiments only perform comparison on ImageNet dataset. Though this dataset has a reasonable size, however, as in many cases, different datasets can bias the performance of different models. To strengthen the results, could the experiments be conducted on multiple datasets as in (Tschannen et al.(2018))?

2. The proposed method is only designed for MobileNets. Is it possible to apply the hybrid filter banks to other neural network architecture?


**Experience Assessment:**

I have read many papers in this area.

**Review Assessment: Checking Correctness Of Derivations And Theory:**

I assessed the sensibility of the derivations and theory.

**Review Assessment: Checking Correctness Of Experiments:**

I assessed the sensibility of the experiments.

**Review Assessment: Thoroughness In Paper Reading:**

I read the paper at least twice and used my best judgement in assessing the paper.

---

> ### Author Response · Authors · 2019-11-15
> **Response to Review #1**
>
> We thank the reviewer for the thoughtful feedback. Please find the responses inline below.
>
> 1. This paper is incremental in nature, with a natural generalization of (Tschannen et al.(2018)).
>
> While our work can be considered an extension of the state-of-the-art StrasseNets technique, its key contributions are as follows:
> (a)	Observation of the difficulty with quantizing depthwise separable convolutional layers (dominated by 1x1 pointwise convolutions) using state-of-the-art techniques
> (b)	Observation of the variance in the sensitivity (listed below) of filters to ternary quantization and subsequent exploitation of this to develop hybrid filter banks to quantize already highly optimized 1x1 pointwise layers of MobileNets architecture.
> (1) Different sensitivity of individual filters to quantization. We empirically showed evidence of that in Figure 1(a)
> (2) Different sensitivity for groups of filters under StrassenNets quantization. We gave a mathematical evidence/proof of that in Figure 1(b) and later in Appendix A.
> (c)	Furthermore, we believe that ~2x compression in model size and 28% saving in energy with no degradation in inference throughput or accuracy is not an incremental contribution for an already compact and compute-efficient architecture like MobileNets.  This paper provides the new best-in-class result for this task.
>
>
> 2. For this kind of paper, I would like to see a more complete set of empirical results. However, The experiments only perform comparison on ImageNet dataset. Though this dataset has a reasonable size, however, as in many cases, different datasets can bias the performance of different models. To strengthen the results, could the experiments be conducted on multiple datasets as in (Tschannen et al.(2018))? The proposed method is only designed for MobileNets. Is it possible to apply the hybrid filter banks to other neural network.
>
>
> Please see answer to Q1 of Review #3 for the additional data points we’ve collected in response to these rebuttals.   We will add these performance results for ResNet, MobileNetV2 on the ImageNet dataset in the final version.

---

### Official Review · AnonReviewer3 · 2019-10-26
**Official Blind Review #3**

**Rating:** 3

**Review:**

The paper presents a quantization method that generates per-layer hybrid filter banks consisting of full-precision and ternary weight filters for MobileNets.

Strength:
(1)	The paper proposes to only quantize easy-to-quantize weight filters of a network layer to ternary values while also preserving the representational ability of the overall network by relying on few full-precision difficult-to-quantize weight filters.
(2)	The proposed method maintains a good balance between overall computational costs and predictive performance of the overall network. Experimental results show that the proposed hybrid filter banks for MobileNets achieves savings in energy and reduction in model size while preserving comparable accuracy.
(3)	The description is clear in general.

Weakness:
(1)	Though the paper claims that recent works on binary/ternary quantization either do not demonstrate their potential to quantize MobileNets on ImageNet dataset or incur modest to significant drop in accuracy while quantizing MobileNets with 4-6-bit weights, it may worth comparing to the methods that achieved start-of-art results on other datasets to demonstrate the efficiency of the proposed method.
(2)	Figure 1 and Figure 2 is a little blurry.
(3).  How is about the performance compared to latest work? Is it possible to apply current framework to MobileNetV2 ? If can, what's performance?


**Experience Assessment:**

I have published one or two papers in this area.

**Review Assessment: Checking Correctness Of Derivations And Theory:**

I assessed the sensibility of the derivations and theory.

**Review Assessment: Checking Correctness Of Experiments:**

I assessed the sensibility of the experiments.

**Review Assessment: Thoroughness In Paper Reading:**

I read the paper at least twice and used my best judgement in assessing the paper.

---

> ### Author Response · Authors · 2019-11-15
> **Response to Review #3**
>
> We thank the reviewer for the thoughtful feedback. Please find the responses inline below.
>
> 1. Comparison to prior work:
>
> (a)	We haven’t found a work that attempt to quantize the weights of MobileNet to ternary or binary values for either CIFAR10, ImageNet or any other dataset. Table 2 in Page 13 of reference [1] further supports this claim. Prior works on ternary quantization [1,2,3,4] primarily evaluate the ResNet architecture on different datasets. To the best of our knowledge, the hybrid filter banks proposed in this work is a first step towards quantizing the already compute-efficient MobileNets architecture to ternary values with a negligible loss in accuracy on a large-scale dataset, such as ImageNet.
> (b)	To address your concern, we have since evaluated the ResNet-20 architecture with hybrid filter banks on the CIFAR-10 dataset to demonstrate its efficacy over other state-of-the-art ternary quantization techniques and its generalizability to other neural network architectures, especially to architectures dominated with 3x3 convolutional layers. ResNet-20 has 19 3x3 convolutional layers. Hybrid filter banks for ResNet-20 consistently achieves a better accuracy for different hidden layer widths when compared to the accuracy reported in StrassenNets [5] while ensuring improvement or no degradation in inference throughput. For example, the Hybrid ResNet-20 with the Alpha = 0.25 and the r = 0.75*cout configuration achieves an accuracy of 91.55% when compared to the accuracy 90.62% observed by StrassenNets with r = 0.75*cout. The accuracy of full-precision ResNet-20 is 92.1. All other configurations consistently outperform the state-of-the-art StrassenNets, demonstrating the generalizability and effectiveness of hybrid filter banks to 3x3 convolutional layers. We will add the performance results of hybrid filter banks for ResNet-20 on CIFAR-10 in the final version.
>
> Alpha = The fraction of channels generated in an output feature map from the full-precision weight filters
> r = The hidden layer width of a strassenified convolution layer (ternary weight filters)
>
> (c)	In this work, we compare hybrid filter banks against StrassenNets and TWN. We will compare hybrid filter banks to the other ternary quantization techniques (such as, TTQ [6] and ABC-Net which are published before StrassenNets [5]) and add their performance results in the final version. As StrassenNets already showed better performance results over the prior ternary quantization methods [6], we concentrated our effort to demonstrate improvement over StrassenNets for MobileNet-like already compact and compute-efficient architecture on a challenging dataset. Furthermore, we believe that when the prior ternary quantization schemes such as TWN, TTQ, ABC-Net could not preserve the accuracy for over-parameterized and less compute-efficient ResNet (in comparison to MobileNet), we should not expect preservation of accuracy for the already compact and compute-efficient MobileNets for these techniques.
>
> References:
> (1)	Christos Louizos et al., “Relaxed Quantization for Discretized Neural Networks”, ICLR 2019
> (2)	Jiwei Yang et al., “Quantization Networks”, CVPR 2019
> (3)	Ruihao Gong et al., “Differentiable Soft Quantization: Bridging Full-Precision and Low-Bit Neural Networks”, ICCV 2019
> (4)	Sangil Jung et al., “Learning to Quantize Deep Networks by Optimizing Quantization Intervals with Task Loss”, CVPR 2019
> (5)	Michael Tschannen et al., “StrassenNets: Deep learning with a multiplication budget”, ICML 2018
> (6)	Chenzhuo Zhu et al., "Trained Ternary Quantization", ICLR 2017
>
>
> 2. Little blurry figures.
> Thank you for pointing to this out; we will fix that in the final version.
>
>
> 3. Current framework to MobileNetV2?
>
> (a)	Yes, it is absolutely possible to apply the current framework and Hybrid Filter Banks to MobileNetV2. In the limited rebuttal time, we could only evaluate MobileNetV2 on the ImageNet dataset with one initial set of model hyperparameters associated with hybrid filter banks approach. The initial results with MobileNetV2 is promising, incurring as little as 2.62% accuracy loss compared to the uncompressed MobilNetV2 *with only the first set of hyperparameters we chose *.  We believe the small accuracy drop can be bridged with more model hyperparameters (e.g., appropriate division of output channels to be generated from either full-precision or ternary weight filters at each layer, appropriate value of hidden layer width for ternary weight filters, etc.) exploration associated with hybrid filter banks approach
> (b)	Furthermore, due to the rebuttal time constraint and long training time of MobileNetV2 on ImageNet we could not apply knowledge distillation (as exploited by the StrasseNets baseline and our hybrid filter banks, mentioned at the end of Section 3).  Knowledge distillation historically improves accuracy by another 1-2%. We will add the performance results for MobileNetV2 with hybrid filter banks in the final version.

---

### Decision · Program_Chairs · 2019-12-19

**Decision:**

Reject

**Comment:**

The paper presents a quantization method that generates per-layer hybrid filter banks consisting of full-precision and ternary weight filters for MobileNets. The paper is well-written. However, it is incremental. Moreover, empirical results are not convincing enough. Experiments are only performed on ImageNet. Comparison on more datasets and more model architectures should be performed.